

# The impact of irrigation on yield of alfalfa and soil chemical properties of saline-sodic soils

Hongtao Yang[1,2], Fenghua An[2], Fan Yang[2] and Zhichun Wang[2]

[1] University of Chinese Academy of Sciences, Beijing, China
[2] Northeast Insitute of Geography and Agroecology, Chinese Academy of Sciences, Changchun city, Jilin Province, China

## ABSTRACT

**Background:** Forage production in the saline-sodic soil of the western Songnen Plain Northeast China depends on irrigation. Therefore, the water use efficiency (WUE) and soil chemical properties are key factors in the overall forage productivity in this water scarce region. Improving forage yield, WUE, and soil properties under irrigation are very important for food and ecological security in this water-deficient region. Additionally, a suitable irrigation schedule for this region is necessary.

**Methods:** A field experiment was conducted between 2015 and 2018 to evaluate the effects of irrigation on artificial grassland productivity and the changes in soil chemical properties as well as to plan a reliable irrigation schedule for the western Songnen Plain. Eight irrigation treatments were designed, which depended on the three growth stages of alfalfa. The shoot height (SH), the chlorophyll content (SPAD), the dry yield (DM), the ratio of stem to leaves (SLR), the WUE, the changes in the chemical properties of the soil, and precipitation and evaporation were investigated.

**Results:** The SH, DM, WUE, and SLR were significantly increased by irrigation ($P < 0.01$). However, the SPAD resulting from irrigation was not significantly higher than the SPAD of CK (no irrigation) ($P < 0.05$). In addition, the soil chemical properties at the depth of 0–100 cm were significantly decreased by irrigation $P$ (0.05). For example, the soil electrical conductivity, sodium absorption ratio, and total alkalization were reduced 182–345 µS cm$^{-1}$, 8.95–9.00 (mmol$_c$/L)$^{1/2}$, and 3.29–4.65 mmol$_c$ L$^{-1}$ by different irrigation treatments, respectively. Finally, considering the highest WUE of I5 (irrigation at branch stage) (2.50 kg m$^{-3}$), relative high DM of I5 (787.00 g m$^{-2}$), the precipitation, the evaporation, the water resources, and the changes of the soil's chemical properties, 236.50 mm of irrigation water was recommended at the branching stage of alfalfa for the western Songnen Plain, Northeast China.

# INTRODUCTION

Numerous environmental factors are potentially detrimental to plants (*Breusegem et al., 2001*). Soil salinity-sodicity is a major environmental factor which limits the plant growth and productivity in the irrigated sections of arid and semi-arid regions (*Qadir et al., 2001*, *2008*;

Corresponding author
Zhichun Wang,
wangzhichun@iga.ac.cn

*Koca et al., 2007*), including the Songnen Plain, Northeast China, where the precipitation cannot maintain a regular infiltration of water through the soil due to the soil's salinity-sodicity. The growth, development, and differentiation of plants growing on saline-sodic lands are generally limited by soil salinization and alkalization; this leads to a decrease in the productivity of the plants. Irrigation is important to maintain a higher forage yield of alfalfa in arid and semi-arid regions (*Guo et al., 2007*).

Saline-sodic soils cover $5.60 \times 10^8$ ha worldwide (*Tanji, 1990*). Such soils are commonly found in arid and semi-arid regions and are characterized by both a high salt content and excessive $Na^+$, which causes damage to the soil structure and a reduction in soil infiltration rate and fertility (*Qadir & Schubert, 2002*; *Qadir et al., 2005*). It has been reported that the saturated hydraulic conductivity of saline-sodic soils is only 0.02–0.22 mm $d^{-1}$ (*Chi & Wang, 2010*), and the infiltration rate decreases quickly after 10 min of irrigation and tends to zero mm $d^{-1}$ after 15 min of irrigation (*Wang et al., 2004*). In addition, the soil pH of saline-sodic soils ranges from 8.5 to 10.5 with 30% to over 70% of exchangeable sodium percentage (*Li, Wang & Chi, 2006*). Today, water shortages and soil salinity-sodicity are considered to be the greatest factors that limit plant survival and growth (*Bandeoğlu et al., 2004*; *Shi & Wang, 2005*). Moreover, the movement of salts follows the soil water flow, which causes a water deficit or low effectiveness of the soil water.

Alfalfa (*Medicago sativa* L.) is a C3 plant and one of the most important perennial legume forages around the world because of its good nutritional forage quality for livestock husbandry and its ability to improve soil fertility (*Guo et al., 2005*; *Dincă et al., 2017*). However, it is generally accepted that alfalfa is a high water requirement species compared with other crops due to its high yield and long growing season (*Bauder, Jacobsen & Lanier, 1992*). Numerous studies have indicated that the seasonal evapotranspiration of alfalfa is approximately 700–1,600 mm depending on the climate and growing period (*Sahin & Hanay, 1996*; *FAO, 2002*). The goals of food security and food sufficiency in the next decades for the western Songnen Plain in northeastern China are linked to the increased availability of irrigation water and high soil fertility. This will increase the pressure on the natural resources of soil and water, considering that the region is characterized by water scarcity and a wide area of saline-sodic soils. Water shortage and soil salinity can be alleviated by enhancing the water use efficiency (WUE) and ameliorating the saline-sodic soils. The Songnen Plain is challenged by a higher rate of evaporation reaching 1,700–1,900 mm compared to an annual precipitation of 370–400 mm, 80% of which occurs in July and August (*Chi & Wang, 2010*). There is a significant difference between the annual precipitation and the water demand of alfalfa. Thus, irrigation is crucial for alfalfa to gain maximum forage yield in this region.

Irrigation is widely used to maintain a higher forage yield of alfalfa in arid and semi-arid regions (*Guo et al., 2007*). However, it is difficult to schedule the irrigation since it would require complex economic and environmental analysis to determine the optimal irrigation operation. Previous studies on artificial grasslands have mainly focused on the comparison of irrigation modes and WUE of alfalfa (*Grimes, Wiley & Sheesley, 1992*; *Estill et al., 1993*; *Potters et al., 2007*; *Kuslu et al., 2010*; *Singh, Kundu & Bandyopadhyay, 2010*). However, there is lack of information on the effects of different irrigation schedules
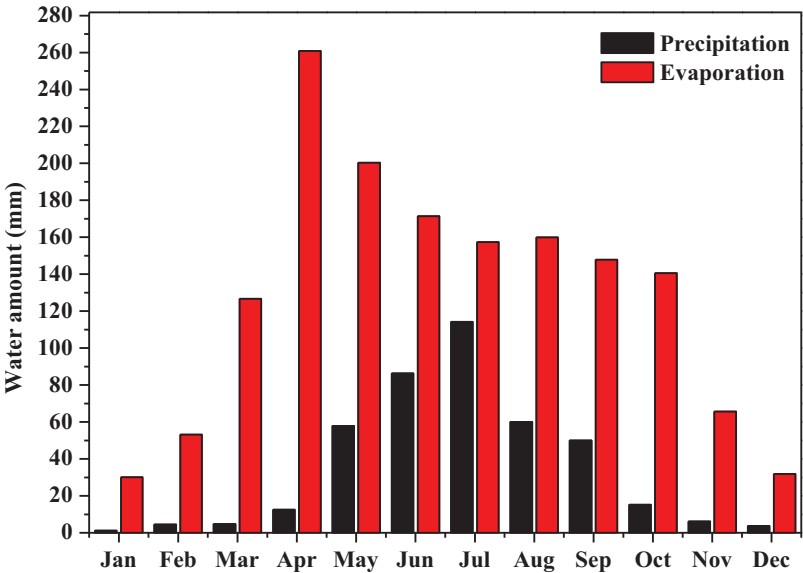

**Figure 1 The monthly precipitation and evaporation of the study site (recorded from 2008.1.1 to 2018.7.31).** This figure showed the monthly precipitation and evaporation of the study site from 2008.1.1 to 2018.7.31.            

on artificial grassland productivity, saline-sodic soil chemical property changes, and suitable water management for the western Songnen Plain, Northeast China.

The objectives of this study were to answer the following questions: (1) which irrigation schedule would be optimal for promoting the productivity of artificial grassland of alfalfa in the western Songnen Plain? (2) How would the irrigation schedules affect the yield of alfalfa and the soil chemical properties?

# MATERIALS AND METHODS

## Study site and plant material

The experiment site (N 45°34′36″–45°34′29″, E 123°2′24″–123°2′47″) was located 15 km east of Baicheng City, Jilin Province, China. The climate of the study site is a temperate continental monsoon climate with an annual precipitation of 370–400 mm and the annual evaporation more than 1,500 mm (Figure 1, data from the Meteorological Bureau of Baicheng City, Jilin Province).

The field experiment was conducted from September 2015 to July 2018 on a saline-alkali artificial grassland, which was established in the spring of 2009 by incorporating 20 cm of sandy soil into a degraded natural grassland. The Alfalfa cultivar of Gongnong No. 1 of *M. sativa* L. cv. was used as the plant material in this experiment and the alfalfa was harvested in July and October every year. The irrigation water was supplied from a local well (Depth > 100 m, pH = 7.5, electrical conductivity (EC) = 50 μS cm$^{-1}$).

The backgrounds of the physico-chemical properties of the soil (at depth of 0–100 cm) are shown in Table 1.

The average soil bulk density (0–100) was 1.59 g cm$^{-3}$ and there was a wide range for the soil pH (8.52–10.45). The soil EC averaged ≥200 μS cm$^{-1}$. The soil at the depth of 0–20 cm
**Table 1 The backgrounds of soil physico-chemical properties.**

| Depth (cm) | Parameter | Average | Maximum | Minimum |
|---|---|---|---|---|
| 0–100 | pH | 9.33 ± 0.09 | 10.45 | 8.52 |
| | EC ($\mu S\ cm^{-1}$) | 330.18 ± 33.18 | 886.00 | 69.60 |
| | SAR (($mmol_c/L$)$^{1/2}$) | 7.65 ± 1.65 | 27.41 | 0.27 |
| | TA ($mmol_c\ L^{-1}$) | 4.35 ± 0.73 | 13.60 | 1.40 |
| | Bulk density (g $cm^{-3}$) | 1.59 ± 0.01 | 1.75 | 1.40 |
| | Field capacity (%) | 37.07 ± 0.44 | 37.07 | 23.48 |
| | Porosity (%) | 40.12 ± 0.36 | 47.29 | 33.81 |
| 0–20 | Sand (%, >0.02 mm) | 84.44 | | |
| | Silt (%, 0.002–0.02) | 10.68 | | |
| | Clay (%, <0.002 mm) | 4.87 | | |
| 20–100 | Sand (%, >0.02 mm) | 44.13 | | |
| | Silt (%, 0.002–0.02) | 44.12 | | |
| | Clay (%, <0.002 mm) | 11.75 | | |

**Note:**
EC, electrical conductivity; SAR, sodium absorption ratio; TA, total alkalinity.

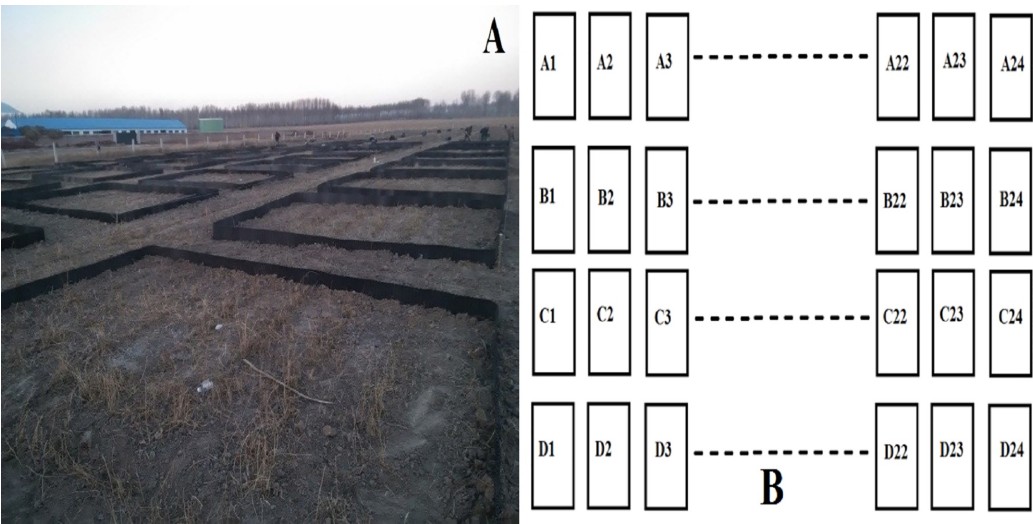

**Figure 2 The distribution and area of experiment plots.** (A) Distribution and area of the irrigation plots in this study; (B) the arrangement and total number of irrigation plots.

and 20–100 cm were classified as sandy soil and loam, respectively, according to the international soil system.

## Experimental design

The spatial distribution of the experiment plots followed a randomized block design (Fig. 2). Each plot was 5 × 4 m = 20 m$^2$.

The alfalfa on the western Songnen Plain was harvested at flower stage, thus, there were eight irrigation treatments based on the three growing stages of alfalfa with twelve replications (Table 2).

**Table 2 The experiment design.**

| Treatments | C (%) | Irrigation amount (mm) | Irrigation time | Irrigation mode | Replications | ET$_1$ | ET$_2$ | ET$_3$ |
|---|---|---|---|---|---|---|---|---|
| I1 | 60 | 236.50 × 2 = 473.00 | R + B | Flood irrigation | 12 | 1,176.12 | 1,248.76 | 424.15 |
| I2 | 60 | 236.50 × 2 = 473.00 | B + F | | 12 | 1,176.12 | 1,248.76 | 424.15 |
| I3 | 60 | 236.50 × 2 = 473.00 | R + F | | 12 | 1,176.12 | 1,248.76 | 424.15 |
| I4 | 30 | 236.50 × 1 = 236.50 | R | | 12 | 1,176.12 | 1,248.76 | 424.15 |
| I5 | 30 | 236.50 × 1 = 236.50 | B | | 12 | 1,176.12 | 1,248.76 | 424.15 |
| I6 | 30 | 236.50 × 1 = 236.50 | F | | 12 | 1,176.12 | 1,248.76 | 424.15 |
| I7 | 90 | 236.50 × 3 = 709.50 | R + B + F | | 12 | 1,176.12 | 1,248.76 | 424.15 |
| CK | 0 | 0 | – | Rain-fed | 12 | 1,176.12 | 1,248.76 | 424.15 |
| Total | – | – | – | – | 96 | – | – | – |

Note:
C, designed irrigation coefficient; R, regreen stage, B, branch sage, F, flower stage; ET$_1$, the mean reference ET for alfalfa regreen stage, mm; ET$_2$, the mean reference ET for alfalfa branch stage, mm; ET$_3$, the mean reference ET for alfalfa flower stage, mm.

The irrigation water amount (I, mm) was calculated as following:

$$I = C \times \mathrm{ET_c} \tag{1}$$

$$\mathrm{ET_c} = \mathrm{ET_0} \times K_c = P + I - D - R \pm \Delta W \tag{2}$$

$$\mathrm{ET_0} = \frac{0.408\Delta(Rn - G) + \gamma \frac{900}{T+273}\mu_2(e_s - e_a)}{\Delta + \gamma(1 + 0.34\mu_2)} \tag{3}$$

ET$_c$, alfalfa evapotranspiration or crop water use, mm; C, design irrigation coefficient, 30%, 60%, 90%, and 0% in this study; $K_c$, crop coefficient, 0.83 (*Van der Gulik, 2001*); ET$_0$, reference ET for alfalfa (mm), calculated by the Penman–Monteith method (*Allen et al., 1998*); Rn, net radiation, MJ m$^{-2}$ d$^{-1}$; G, soil heat flux density, MJ m$^{-2}$ d$^{-1}$; T, mean daily air temperature at two m height, °C; $\mu_2$, wind speed at two m height, m s$^{-1}$; $e_s$, saturator vapor pressure, kPa; $e_a$, the actual vapor pressure, kPa; $\Delta$, slope of the vapor pressure curve, kPa °C$^{-1}$; $\gamma$, psychometric constant, kPa °C$^{-1}$.

## Data collection

The relative chlorophyll content (SPAD) in each plant was determined using a SPAD-502 Chlorophyll Meter (Minolta Co. Ltd., Osska, Japan). The SPAD was measured every week following the beginning of the regreen stage of alfalfa. The SPAD of each alfalfa was measured five times and then the mean values of SPAD were calculated. Five alfalfa stems were selected randomly from each plot for measuring the shoot height (SH) using a ruler from the base of stem to the end.

Dry yields (DM) were measured by harvesting a 1.00 × 1.00 m quadrat from the center of each plot when the alfalfa reached 50-flower. Samples were air-dried in a repository. The DM was calculated by the 1st and the 2nd harvest of alfalfa DM.

Each subsample of alfalfa was separated into the leaf and stem fraction for measuring the ratio of stem to leaves (SLR). The SLR was calculated as the average of the 1st and 2nd harvest of alfalfa.

The soil water balance and $ET_c$ were calculated using following equation (*Carter & Sheaffer, 1983*):

$$ET_c = P + I - D - R \pm \Delta W \tag{4}$$

$ET_c$, evapotranspiration, mm; $P$, precipitation, mm; $I$, irrigation amount, mm; $D$, downward drainage out of root zone, mm; $R$, surface runoff, mm; $\Delta W$, change in soil water storage, mm.

Drainage ($D$) below the root zone was assumed to be zero since water applied with each irrigation was less than or equal to the water deficit in the one m soil profile of the fully irrigated treatment. Runoff ($R$) was considered zero because the experimental plots were surrounded by 0.15 m high and 20 m long plastic levees around its perimeter and the basin were meticulously prepared to be level.

The meteorological data from 2015 to 2018, including precipitation ($P$), were collected by the meteorological station (HOBO U30 Weather station; Onset Computer Corporation, Bourne, MA, USA). The actual amount of irrigation ($I$, mm) was recorded by the water meter set up in each plot. Neutron probe tubes were installed for measuring the $\Delta W$ weekly at the depth of 0–100 cm.

The irrigation should be slight and applied frequently at every growth stage to avoid water-logging the plot. WUE (kg m$^{-3}$) was calculated according to Eq. (2) (*Ojedaa et al., 2018*):

$$WUE = \frac{Y}{ET_c} \tag{5}$$

$Y$, yield, g m$^{-2}$.

The water table significantly declined following the extreme drought from 2015 to 2017, and large areas of new paddy fields were developed nearby. Thus, hydraulic pressure was too low to support all the plots equally. This may have caused insignificant differences in the actual water consumption among the different irrigation treatments.

The alfalfa water sensitive indexes of different growth stages were calculated as (*Jensen, 1968*):

$$\frac{Y}{Y_m} = \prod_{i=1}^{n} \left( \frac{ET_i}{ET_{mi}} \right)^{\lambda_i} \tag{6}$$

$n$, stage number of alfalfa; $i$, serial number of alfalfa growth stages; $Y$, actual yield, kg ha$^{-1}$; $Y_m$, maximum or potential grain yield with water not limiting production, kg ha$^{-1}$; $ET_i$, actual evapotranspiration in growth stage $i$, mm; $ET_{mi}$, maximum or potential evapotranspiration in growth stage $i$, mm; $\lambda_i$, water sensitive index (WSI) in growth stage $i$.

Soil samples were collected with a soil auger at a depth of 0–100 cm in September 2015 and October 2017 in order to measure for soil chemical properties, such as soil pH, soil EC, sodium absorption ratio (SAR), and total alkalization (TA).

The collected soil samples were air-dried, mixed, sieved through a two mm sieve, and then analyzed according to the methods described by the *United States Department of Agriculture (USDA) (1954)*. Soil pH and EC were measured in a soil-water suspension with

a 1:5 soil/water ratio, using a pH and EC meter (LeiCi Co. Ltd., Shanghai, China, and specifications of the glass electrodes were DJS-1C and PHSJ-3F, respectively). The $K^+$ and $Na^+$ concentrations from the soil-water extracts were analyzed by using flame photometry (Shanghai Precision Co. Ltd., Shanghai, China), whereas the concentrations of $Ca^{2+}$ and $Mg^{2+}$ were determined by atomic absorption spectrometry (Haiguang GGX605; Shanghai, China). The SAR and TA were calculated by the equations below (*Wang et al., 2018*).

$$SAR = \frac{[Na^+]}{\sqrt{[Ca^{2+} + Mg^{2+}]/2}} \tag{7}$$

$$TA = [CO_3{}^{2-}] + [HCO_3^-] \tag{8}$$

The differences of soil pH, EC, SAR, and TA at 2015 and 2017 were calculated for representing the changes in the soil chemical properties.

$$\Delta pH = pH_{2015} - pH_{2017} \tag{9}$$

$$\Delta EC = EC_{2015} - EC_{2017} \tag{10}$$

$$\Delta SAR = SAR_{2015} - SAR_{2017} \tag{11}$$

$$\Delta TA = TA_{2015} - TA_{2017} \tag{12}$$

## Data analysis

For each soil sample, one-way ANOVA was performed on SLR, DM, SH, SPAD, and WUE using SPSS ($P < 0.05$) (Version 20.0; IBM, Armonk, NY, USA). Multiple linear regression was conducted to determine the relationship between the actual alfalfa yield and the predictable alfalfa yield.

Origin 8.0 (OriginLab Corporation, Northampton, MA, USA) was employed for the figuring.

## RESULTS

### Precipitation and evaporation during the growing season

The results of the monthly precipitation and evaporation during the alfalfa growth periods from 2015 to 2018 (during the April and October) were shown in Fig. 2.

The total precipitation and evaporation during the alfalfa growth season were 399.25 and 1,020.01 mm, respectively (Fig. 3). The highest monthly precipitation was 91.75 mm, which was observed in July, and the lowest monthly precipitation was 7.15 mm, which was observed in April. The highest monthly evaporation was 205.60 mm (April) which was double the monthly evaporation of October (101.67 mm).

### The bio-characteristics of alfalfa under different treatments

The SH (cm), SPAD, DM (g m$^{-2}$), and the SLR of different treatments were shown in the Table 3.

It was clear that the alfalfa DM, SLR, and SH were significantly ($P < 0.05$) influenced by irrigation in the western Songnen Plain. The DMs of the irrigated plots were 455.26–1,393.86% heavier than the DM of CK. The SH of irrigation treatments were

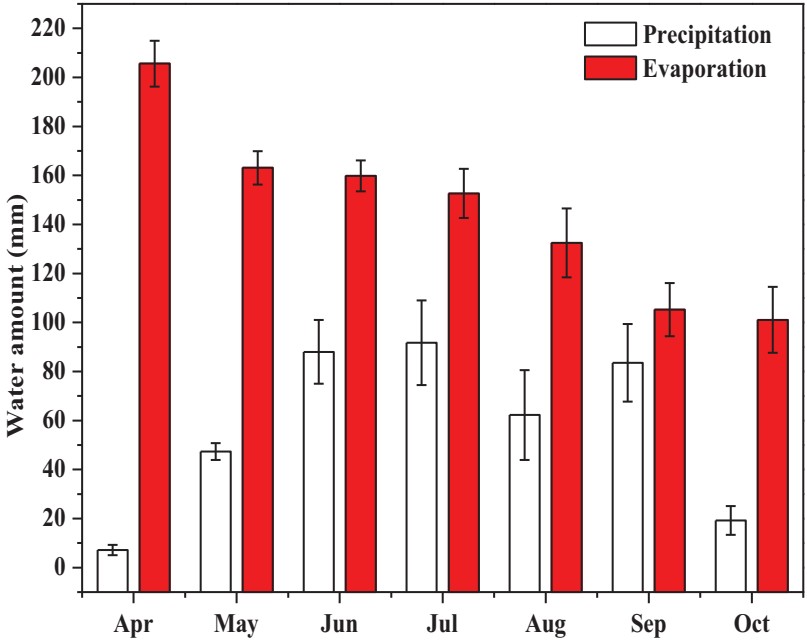

**Figure 3 The monthly precipitation and evaporation during the growing season of alfalfa (*P*, precipitation, mm; *E*, evaporation, mm).** The monthly precipitation and evaporation during the alfalfa growing season (from April to October) are shown.

**Table 3 The DM, SH, SPAD, and SLR of alfalfa under different treatments.**

|  | I1 | I2 | I3 | I4 | I5 | I6 | I7 | CK |
|---|---|---|---|---|---|---|---|---|
| SLR | 2.06 ± 0.09[a] | 2.24 ± 0.08[a] | 2.17 ± 0.06[a] | 2.08 ± 0.07[a] | 1.87 ± 0.03[a] | 2.24 ± 0.06[a] | 2.62 ± 0.07[a] | 2.14 ± 0.01[a] |
| DM | 953 ± 34.20[b] | 980 ± 52.92[b] | 873 ± 34.64[b] | 647 ± 94.04[c] | 787 ± 59.25[c] | 633 ± 35.28[c] | 1,703 ± 97.01[a] | 114 ± 19.16[d] |
| SH | 109.78 ± 3.61[bc] | 110.22 ± 1.79[bc] | 110 ± 4.67[bc] | 98.67 ± 5.35[b] | 115.44 ± 1.93[ab] | 104 ± 3.86[b] | 123.78 ± 2.41[a] | 55.11 ± 1.44[e] |
| SPAD | 58.21 ± 3.04[a] | 58.89 ± 1.49[a] | 60.21 ± 3.60[a] | 64.28 ± 2.20[a] | 59.71 ± 4.64[a] | 61.30 ± 3.04[a] | 64.87 ± 0.98[a] | 53.33 ± 5.71[a] |

**Note:**
SLR, mass ratio of stems to leaves; DM, dry yield, g m$^{-2}$; SH, shoot height of alfalfa, cm; I1, irrigate at regreen and branch stages; I2, irrigate at branch and anthesis stages; I3, irrigate at regreen and anthesis stages; I4, irrigate at regreen stage; I5, irrigate at branch stage; I6, irrigate at anthesis stage; I7, irrigate at regreen, branch and anthesis stages; CK, no irrigation. Lowercase letters indicate that the parameters are significantly different at $P < 0.05$.

79.04–124.61% higher than the SH of CK. In addition, SLRs of irrigation treatments were 1.87% higher than the SLR of CK. Furthermore, irrigation treatment enhanced the chlorophyll concentration more than the CK; however, there were no significant differences ($P > 0.05$).

The DMs of alfalfa irrigated at only one growth stage was significantly ($P < 0.05$) lower than the DMs of alfalfa irrigated at two growth stages. The DM of I2 reached up to 980 g m$^{-2}$, which was the highest yield among the deficit irrigation treatments, which was followed by I1 (953.33 g m$^{-2}$). The highest DM of alfalfa irrigated at one growth stage was observed at the I5 treatment (786.67 g m$^{-2}$). Evidence observed in this study showed that the effect of irrigation treatments on SLR was not significant ($P > 0.05$). The SLRs of I1, I4, and I5 were decreased and the SLRs of I2, I3, I6, and I7 were increased by irrigation. The highest SLR was 2.62, which was observed at full irrigation treatment (I7) and

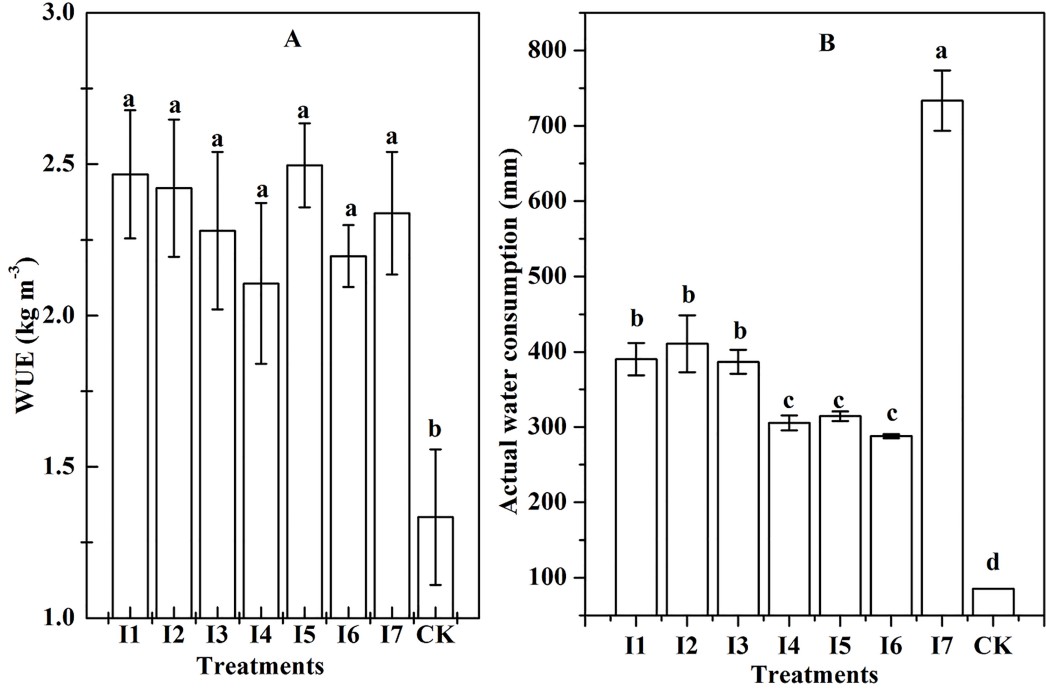

**Figure 4 The WUE and water consumption of different treatments ($P < 0.05$).** (A) Result of the water use efficiency (WUE) of different irrigation treatments; (B) the actual water consumption of the eight treatments. Lowercase letters indicate the results of ANOVA at $\alpha = 0.05$; in addition, the double asterisks show that the confidence interval is $\alpha = 0.05$.

then followed by SLR of I2 (2.24) and I2 (2.2.4). The SHs of eight irrigation treatments ranged from 55.11 to 123.78 cm. Results of SH showed that I7, which was irrigated at regreen, branch, and flower stages, performed the best on the increase of the alfalfa SH, followed by the I2 and I6 treatments. The values of SPAD increased from 53.33 to 64.87 among the eight irrigation treatments. Maximum SPAD (64.87) was observed from the I7 treatment, and the minimum SPAD (53.33) from CK.

## The actual water consumption and water use efficiency

The total amount of water applied to alfalfa ranged from 85.00 to 755.00 mm in different irrigation treatments (Fig. 4). The actual water consumption of I1, I2, and I3 were 390.18, 410.82, and 386.59, respectively. Additionally, the actual water consumption of I4, I5, and I6 was 305.46, 314.48, and 288.08 mm, respectively. Furthermore, the I7 consumed 733.00 mm of water. The lowest water consumption was observed in the CK treatment (85.49 mm).

The WUE of seven treatments with irrigation ranged from 1.33 to 2.50 kg m$^{-3}$ and were significantly higher than the WUE of CK ($P < 0.05$), however, no differences among the WUEs of irrigation treatments were found ($P < 0.05$). The lowest WUE (1.33 kg m$^{-3}$) was observed at CK. There were no significant differences in WUE between the I1 (2.47 kg m$^{-3}$), I2 (2.42 kg m$^{-3}$), and I3 (2.28 kg m$^{-3}$), which were irrigated at two growth stages of alfalfa. Compared to I4 and I6, which were irrigated at one growth stage of alfalfa, I5 has reached the highest WUE (2.50 kg m$^{-3}$). This might imply that irrigation at the branch stage has an important influence on the WUE of alfalfa.

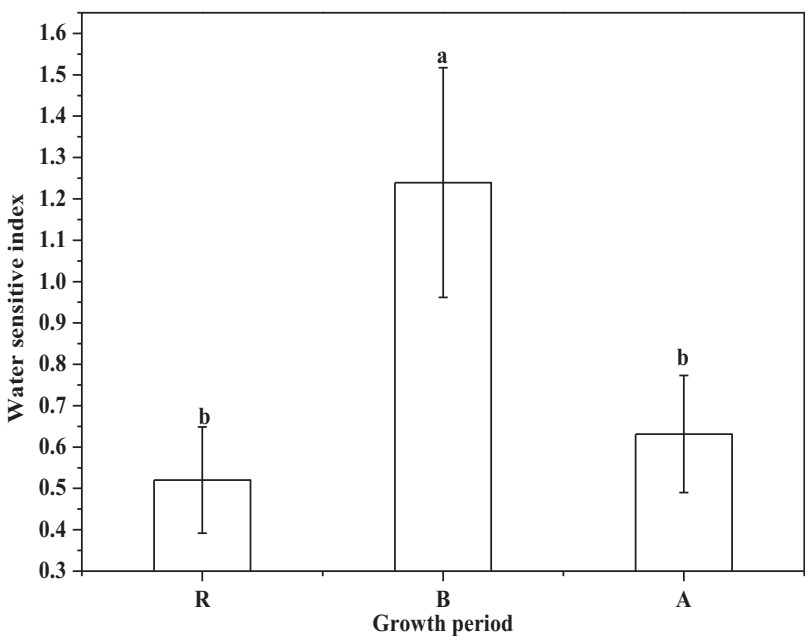

**Figure 5** **The water sensitive index of alfalfa (R, regreen period; B, branch period; A, flower period).**
The water sensitive index of alfalfa in regreen, branch, and flower stage. Lowercase letters indicate the
results of ANOVA at α = 0.05.

## Irrigation schedule for western Songnen Plain

The water insensitive indexes of alfalfa at the regreen, branch, and flower stages were
shown in Fig. 5 below.

The Fig. 5 showed that the WSI of the branch period was significantly higher ($P < 0.05$)
than the WSIs of the regreen and flower periods. In addition, there was no significant
($P < 0.05$) difference between the WSI of regreen and flower.

The water production function of alfalfa in the western Songnen Plain was determined
with $\lambda_1$, $\lambda_2$ and $\lambda_3$ these was shown in Fig. 5. Then the water production function of alfalfa
in the Songnen Plain was displayed in Eq. (13).

$$\frac{Y}{Y_m} = \left(\frac{ET_1}{ET_{m1}}\right)^{0.5202} \times \left(\frac{ET_2}{ET_{m2}}\right)^{1.2393} \times \left(\frac{ET_3}{ET_{m3}}\right)^{0.6313} \tag{13}$$

The determination correlation coefficient ($r = 0.71$) between the simulated alfalfa DM
and the amount of irrigation water was 0.71, which was in the critical coefficient between
the simulated alfalfa DM and the amount of irrigation water (0.30–0.93). Thus, the
Jensen model was reliable for the western Songnen Plain, Northeast China for irrigation
scheduling. The simulated yield obtain from the Jensen model of the western Songnen
Plain was lower than the actual yield (Fig. 6). However, the simulated alfalfa DM
would be closer to the actual alfalfa DM when the amount of irrigation water reached
at 150.00–350.00 mm.

The relationship between the alfalfa yield and the amount of irrigation on the
artificial grassland in the western Songnen Plain can be simulated with the cubic curve

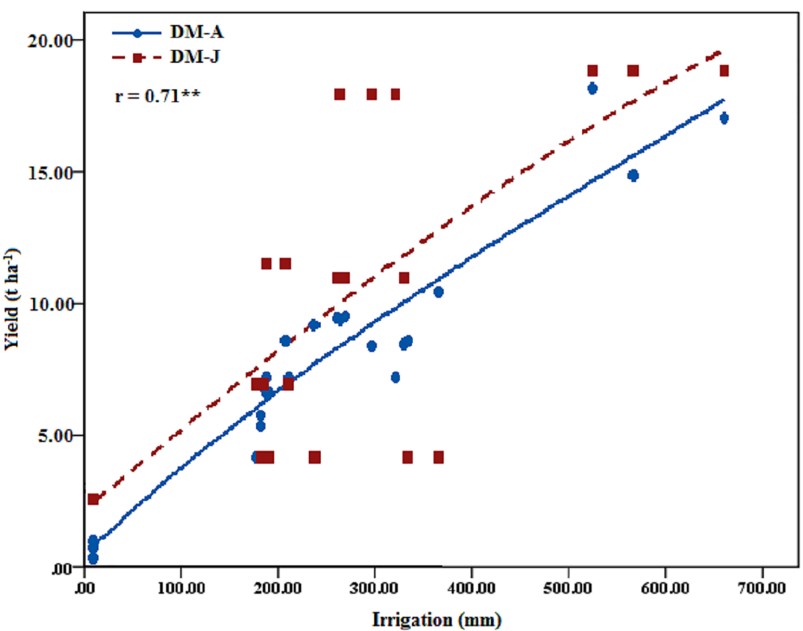

**Figure 6 Actual and simulated of alfalfa yield and the relations to the irrigation amount (DM-J, alfalfa yield simulated by Jensen Model, $t$ ha$^{-1}$; DM-A, actual alfalfa yield, $t$ ha$^{-1}$).** The actual alfalfa yield and simulated yield by Jensen Model were correlated to the amount of irrigation, respectively, and the results are shown. **show that the confidence interval is $\alpha = 0.05$.

($y = 125.188 + 3.449\,x + 0.002\,x^2 - 10^{-6}\,x^3$, $650 > x > 0$, $R^2 = 0.91$, $F = 65.61 > F_{0.01}\,(3, 20) = 4.94$) (Fig. 7). Alfalfa yield could rapid increase coupled with the amount of irrigation increase when less than 650 mm water was supplied.

Based on the data above, 236.50 mm of irrigation water was recommended at the branch stage of alfalfa for the eastern Songnen Plain, Northeast China.

## Effects of irrigation on soil chemical properties

Results of changes in soil chemical properties after 2-years of irrigation were shown in the figures below.

Irrigation had significant effects on the chemical properties of the soil at different soil depths (0–100 cm, Figs. 8–11). For example, the soil EC, SAR, and TA of irrigation treatments showed greater decreasing effects than CK. During the growing season, the average soil EC of different irrigation treatments decreased 210.21–287.24 µS cm$^{-1}$, and the average soil EC of CK decreased 46.35 µS cm$^{-1}$. The results of soil SAR (average of 0–100 cm) revealed that irrigation reduced 8.10–9.00 (mmol$_c$/L)$^{1/2}$, and on the contrast, the average soil SAR of CK decreased only 0.93 (mmol$_c$/L)$^{1/2}$ from September 2015 to October 2017.

The greatest decrease of soil TA was observed with the I7 irrigation treatment (4.64 mmol$_c$ L$^{-1}$), while the lowest decrease of soil TA was found in the I3 treatment (3.29 mmol$_c$ L$^{-1}$), and the soil TA without irrigation decreased 2.88 mmol$_c$ L$^{-1}$. The effects of irrigation on soil pH were complicated. For instance, the soil pH at a depth of 0–40 cm and 80–100 cm decreased at 0.003–0.80, and a 0.13–0.56 increase of soil pH at the depth 40–60 cm was found.

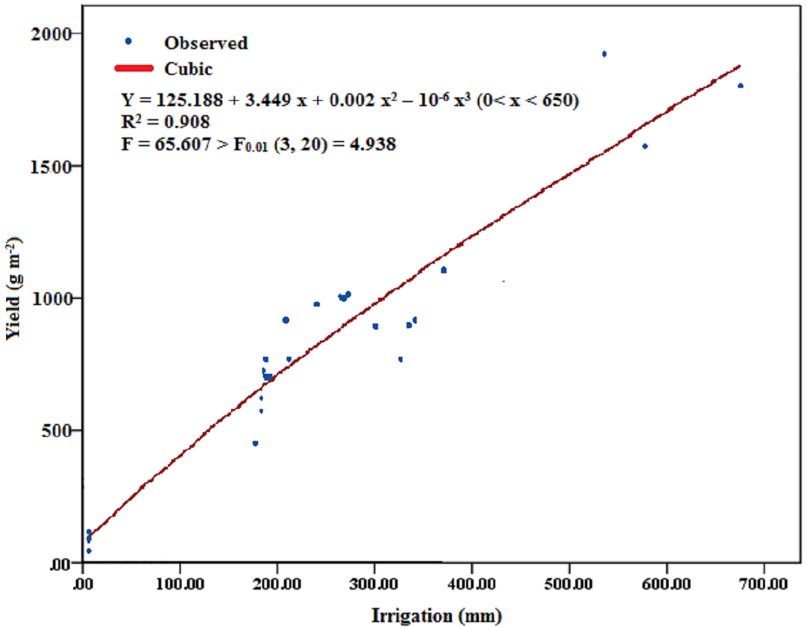

**Figure 7 The relation between amount of irrigation and actual alfalfa yield.** The relation between the actual alfalfa yield and the amount of irrigation was calculated by the curve fitting program in SPSS 20.0.

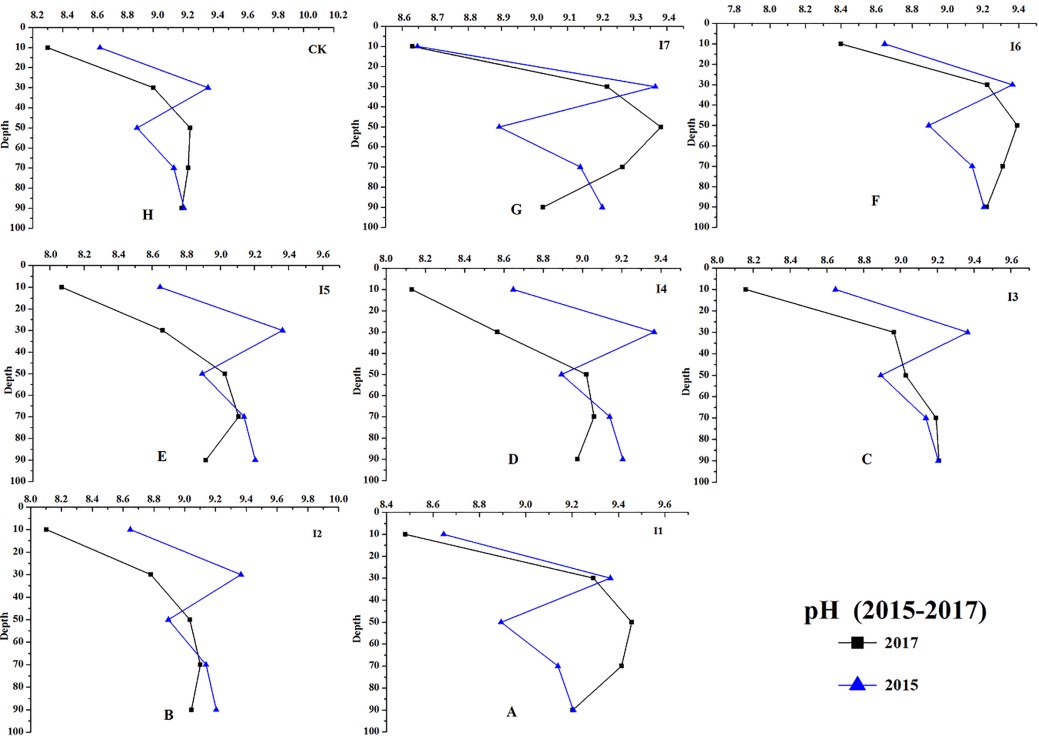

**Figure 8 The soil pH changes of different irrigation treatments at depth of 0–100 cm (from 2015 to 2017).** The changes of soil pH from 2015 to 2017 are shown. (A–H) Treatments of I1, I2, I3, I4, I5, I6, I7, and CK, respectively.

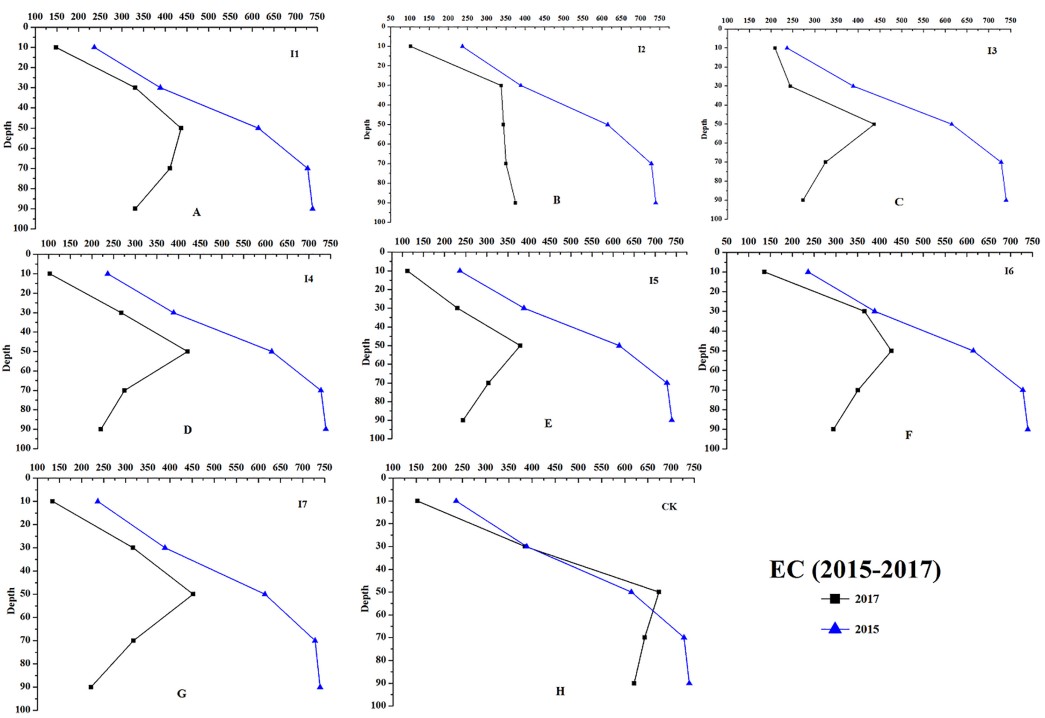

**Figure 9 The soil EC changes of different irrigation treatments at depth of 0–100 cm (from 2015 to 2017).** The soil EC changes at depth of 0–100 cm from 2015 to 2017. (A–H) Treatments of I1, I2, I3, I4, I5, I6, I7, and CK, respectively.

As shown in Fig. 8, I2, I3, I4, and I5 showed a greater decrease on soil pH (0–00 cm, $\Delta$pH = 0.14–0.30) than the other treatments. The soil EC of I4 and I5 decreased 283.89 and 287.24 $\mu$S cm$^{-1}$, which were larger than other treatments (Fig. 9). The soil SAR decreases 8.96 and 9.00 (mmol$_c$/L)$^{1/2}$, which were observed with the I7 and I5 treatments (Fig. 10); the soil TA under the treatments of I5, I6, and I7 ($\Delta$TA = 4.15–4.65 mmol$_c$ L$^{-1}$) showed a greater decrease than the other treatments (Fig. 11).

## DISCUSSION

Alfalfa is a crop with a high water demand (*Bauder, Jacobsen & Lanier, 1992*) and its biomass production depends on SH, leaf area, and leaf number, which can be reduced drastically by a deficit in the soil water (*Saeed & El-Nadi, 1997*). The alfalfa DMs of irrigated treatments in this study ranged from 633.00 to 1,703.00 g m$^{-2}$ and significantly correlated with SH ($r$ = 0.61, $P$ < 0.01), which indicated that water is critical in the biomass production of alfalfa. The high correlation between SH and biomass production found in this study was in agreement with *Davis & Buker (1966)* who reported that 65% of alfalfa yield was determined by SH. The WUE of alfalfa in this study was promoted from 1.33 to 2.50 kg m$^{-3}$, which agreed with previous reports in cooler, more northerly climates (*Stanhill, 1986*; *Bogler & Matches, 1990*; *Grimes, Wiley & Sheesley, 1992*; *Saeed & El-Nadi, 1997*). The highest WUE was observed in the I5 treatment (2.50 kg m$^{-3}$) which was irrigated at the branch stage of alfalfa and then followed by I1 treatment (2.47 kg m$^{-3}$). In addition, results of *Kuslu et al. (2010)* and *Estill et al. (1993)* indicated that the DM and

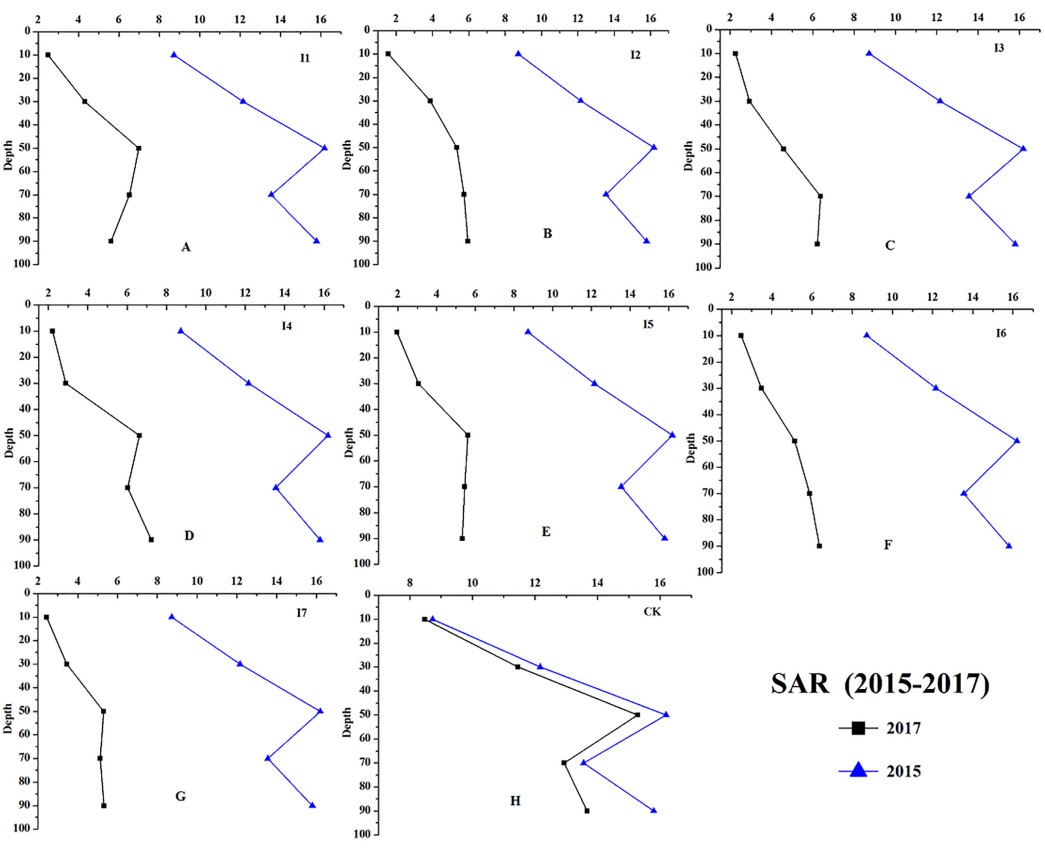

**Figure 10 The soil SAR changes of different irrigation treatments at depth of 0–100 cm (from 2015 to 2017).** The soil SAR changes from 2015 to 2017 at depth of 0–100 cm. (A–H) Treatments of I1, I2, I3, I4, I5, I6, I7, and CK, respectively.

the WUE of alfalfa were decreased by water shortage. Thus, irrigation is a key factor to maintain and promote the WUE and productivity of artificial grassland (*Potters et al., 2007*). We also found a significant positive correlation ($r = 0.42$, $P < 0.05$) between WUE and the amount of irrigation water in this study. Water shortage significantly hindered the plant cell elongation, reduced photosynthesis, interfered with water and nutrient uptake, and changed the plant hormone levels (*Saeed & El-Nadi, 1997*; *Antolín & Sánchez-Díaz, 1993*). Thus, the results of this study indicated that the branch stage is the most important stage for alfalfa in determining the stem density, SH, and the area and amount alfalfa leaf. Moreover, the SLR results showed that the effect of irrigation on SLR was not significant ($P > 0.05$), however, the SLR of CK treatment was decreased compared to I1, I4, and I5, which indicated that the forage quality might increase under irrigation. Furthermore, results of this study showed that there is a difference between precipitation and evaporation in the western Songnen Plain. Therefore, in order to promote the WUE, DM and quality of crops in this region, perennial crops such as alfalfa should be irrigated in a rational schedule.

Plants, through their capacity to assimilate, transport, and evaporate soil water, leave a strong imprint on water and salt dynamics. In addition, the dynamics of soluble salts

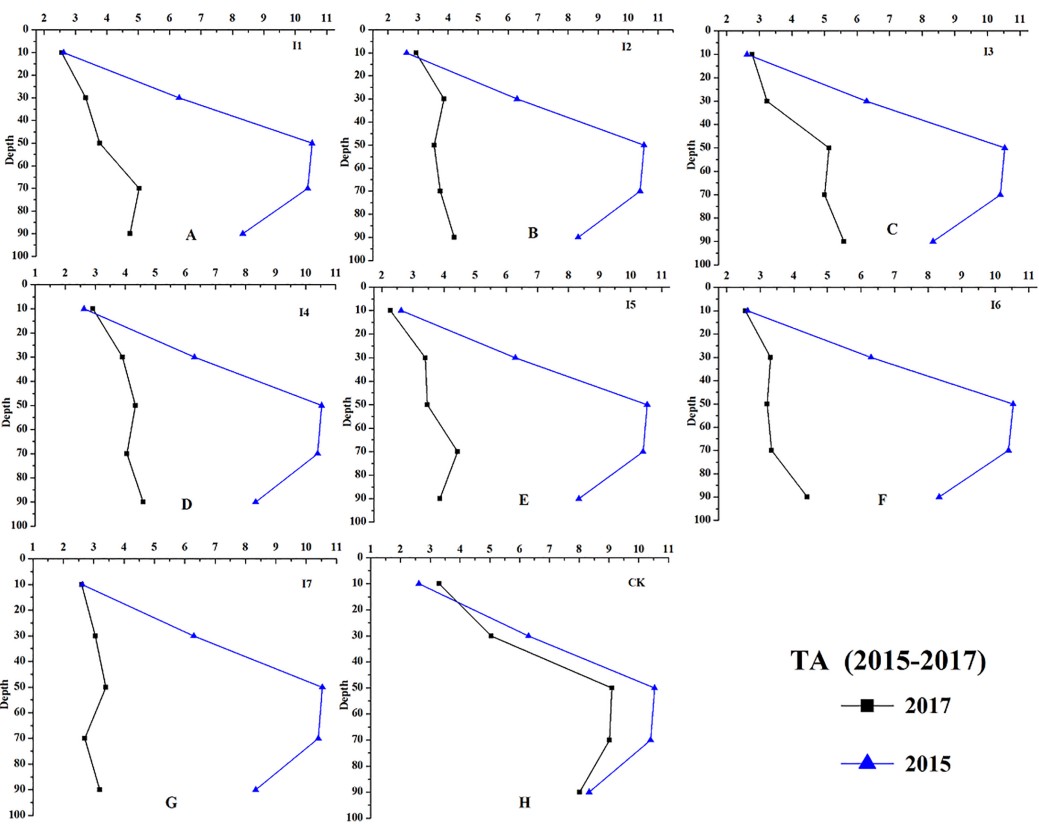

**Figure 11 The soil TA changes of different treatments at depth of 0–100 cm (from 2015 to 2017).** The result of soil TA changes at depth of 0–100 cm was shown. (A–H) Treatments of I1, I2, I3, I4, I5, I6, I7, and CK, respectively.

in soils are significantly linked to water movement through the ecosystem-vadose zone-aquifer continuum (*Nosetto et al., 2007*). While most of the knowledge about the effects of irrigation and vegetation on salt dynamics derive from higher evapotranspiration rates that drive the movement of groundwater and salt, and increased soil macro-porosity enhancing water infiltration and salt flushing. Results of this study indicated that the I2, I5, and I7 treatments showed better effects of de-salinization and de-alkalization than the other irrigation treatments. For instance, the 0.23 and 0.29 of soil pH of I2 and I5 treatments were decreased by irrigation, which was similar to the research of *Dong, Guo & Kong (2001)*. Research of *Cao et al. (2012)* indicated that the soil EC and concentration of $HCO_3^-$ decreased after planting alfalfa for 3 years in the Yingda irrigation district. Many field experiments indicated that the vegetation decreased evaporation and salt accumulation in the surface soil (*Ghaly, 2002*; *Lin, Jia & Zhang, 2003*). Our research also noted that the soil EC of I2, I5, and I7 declined 240.30, 287.25, and 253.10 $\mu S\ cm^{-1}$; additionally, the soil TA of I2, I5, and I7 diminished 3.98, 4.15 and 4.65 $mmol_c\ L^{-1}$, respectively. These results may be attributed to the deep taproot system of alfalfa (*Raiesi, 2007*; *Yong, 2007*). Evapotranspiration decreased with vegetation cover (*Nosetto et al., 2007*). Furthermore, large amounts of salts in the soil were absorbed and moved out of the root zone by the harvested forages, however, there were less salts from irrigation and

precipitation in the soils. These factors may contribute to the chemical property changes in the soil of the western Songnen Plain, Northeast China.

## CONCLUSIONS

Results of this study demonstrated that the DM and WUE of alfalfa in saline-alkali artificial grasslands were promoted from 114.00 to 1,703.00 g m$^{-2}$ and 1.33 to 2.50 kg m$^{-3}$ by irrigation, respectively. In addition, the soil EC, SAR, and TA (depth of 0–100 cm) were decreased by irrigation. Considering the local precipitation, evaporation, water resources, and backgrounds of soil chemical properties, 236.50 mm of irrigation water at the branch stage was recommended for artificial grassland in the western Songnen Plain, Northeast China.

## ABBREVIATIONS

| | |
|---|---|
| $ET_0$ | the reference evapotranspiration, mm |
| $ET_a$ | the actual evapotranspiration, mm |
| $Y$ | actual yield, kg ha$^{-1}$ |
| $Y_m$ | maximum or potential grain yield with water not limiting production, kg ha$^{-1}$ |
| $ET_i$ | actual evapotranspiration in growth stage $i$, mm |
| $ET_{mi}$ | maximum or potential evapotranspiration in growth stage $i$, mm |
| $\lambda_i$ | water sensitive index in growth stage $i$ |
| $I$ | the amount of irrigation water, mm |
| $P$ | precipitation, mm |
| $D$ | the amount of drainage water, mm |
| $R$ | amount of runoff, mm |
| $\Delta W$ | changes in the soil water content, mm |
| DM | dry yield of alfalfa, g m$^{-2}$ |
| SH | shoot height, cm |
| SPAD | the content of chlorophyll (SPAD) |
| SLR | ratio of stem to leaves, % |
| $C$ | design irrigation coefficient |
| $K_c$ | crop coefficient |
| WUE | water use efficiency, kg m$^{-3}$ |
| EC | soil electrical conductivity, μS cm$^{-1}$ |
| SAR | sodium absorption ratio, (mmol$_c$/L)$^{1/2}$ |
| TA | total alkalization, mmol$_c$ L$^{-1}$ |

### Funding

This work was supported by the foundations of the National Natural Science Foundation of China (41571210, 41771250), The Technology Development Project of Jilin province (20180201012SF), The National Key Research and Development program of China

(2016YFC0501200), the National Science and Technology Basic Work of China (2015FY110500) and the National Natural Science Foundation (NSCF, 41701335). The funders had no role in study design, data collection and analysis, decision to publish, or preparation of the manuscript.

## Grant Disclosures

The following grant information was disclosed by the authors:
National Natural Science Foundation of China: 41571210, 41771250.
National key research and development program of China: 2016YFC0501200.
National science and technology basic work of China: 2015FY110500.
National natural science foundation: NSCF, 41701335.

## Competing Interests

The authors declare that they have no competing interests.

## Author Contributions

- Hongtao Yang conceived and designed the experiments, performed the experiments, analyzed the data, contributed reagents/materials/analysis tools, prepared figures and/or tables, authored or reviewed drafts of the paper, approved the final draft, experiment design, draft, data analysis, submission.
- Fenghua An contributed reagents/materials/analysis tools.
- Fan Yang contributed reagents/materials/analysis tools.
- Zhichun Wang contributed reagents/materials/analysis tools.

## Data Availability

The raw data is available in a Supplemental File.

## Supplemental Information

Supplemental information for this article can be found online at http://dx.doi.org/10.7717/peerj.7148#supplemental-information.

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
