# Peer review of "The impact of irrigation on yield of alfalfa and soil chemical properties of saline-sodic soils"

_PeerJ, doi:10.7717/peerj.7148_

## Round 0.1 · original submission · Major Revisions

Please follow closely the reviewers' comments and revise accordingly. The revised copy will be sent out for re-review to the same reviewers.

Reviewer 1 ·

Basic reporting

The article deals with the Alfalfa and soil chemical properties response to irrigation in saline-sodic soil region. The writing is clear enough, however there are many grammatical and structural mistakes in the manuscript and I am not able to highlight/revise all of them. In addition, this paper needs quite a few places for professional editing. So, please revise the manuscript and improve it considerably in terms of these issues.
The literature is suitable for the purpose of the article, but the locations of researchers' first and last name is inappropriate (Line 357, 360, 381...).

Experimental design

There is insufficient detail in the soil sample collection section of the "Research method" section to reproduce the fieldwork and lab work.
How many plots are designed in your study? And the detailed geographic locations of them?You should add a clear statement and/or sketch map in the revision.
Which "soil types" determined your sampling densities? A detailed illustration of soil is necessary. Report soil types according to an international soil system, either the WRB or USDA systems.
You have listed the specifications of the pH and EC meter but not the specifications of the detailed glass electrode, if that's what you've used. What other properties of the soil did you measure? These are required for your conclusions.

Validity of the findings

Data is robust, statistically sound, & controlled.

Additional comments

1. The text is extensive and repetitive in many parts. Thus, I recommend rewrite abstract and introduction. The conclusions can be furtherly improved.
2. Work on text clarification and formatting, improve figures, list of the abbreviations would be useful in this case to easily understand the text.
3. The caption of Figure 1 seems to be wrong. Please double check.
4. The units of X and Y axis are missing in the Figure 7. Please add them.
5. In all the numbers with decimals, please consider to homogenize to only two/three decimal.
6. The extensive and methodical experimentation proves the effectiveness of the selected methods. However, the paper would gain in clarity if the achieved results were correlated with the characteristics of the algorithms.
7. The comprehensive discussion of effects of different irrigation treatments should be added in the revision.
8. Lines 105-106: There were eight irrigation treatments …Need to justify why these different treatments were designed and used specifically.
9. Lines 107-108 The equation numbers were missing. Please add them.
10. Line 109-111 ETc, … calculated by Penman-Monteith method. The origin of essential parameters should be illustrated clearly.
11. Lines 116-117 Five stems will be selected at random from each plot for measuring… which instrument was applied for these measurements.
12. Lines 129-130; 154-157 Please rewrite these sentences, they are difficult to follow.
13. Lines 177-178 The explanation given is hardly to understand.
14. Line 183 “rainfall” should be “precipitation”.
15. Description of Result and Discussion is not very professional. Several sentences in the Result section should be in the Method or in the figure captions. You don't need to say ....are shown in Figure X, simply put (Figure X) next to the sentences that describing the results. Paragraphs in the Discussion should start with a topic sentence instead of describing the methods again.
16. The discussion is weak and unsatisfied, the author should make the best effort to modify this section. I want to see a considerable improvement in this regard.
17. The conclusion was not well stated and written more as an abstract. A clear conclusion needs to be written.
Overall this manuscript adds practical values to agriculture management. I suggested authors to revisit this paper several times to make it more concise based on the above comments and comments from other reviewers.

Reviewer 2 ·

Basic reporting

Title:
- "Alfalfa and soil chemical properties response to irrigation in saline-sodic soil region" uses incorrect English grammar. The response is possessed by the chemical properties, or the irrigation is something to which chemical properties respond. Therefore the title should read as, for instance, "Alfalfa and soil chemical properties respond to irrigation in saline-sodic soil region." Ideally, you would be more specific. The best titles state the results in 18 words or less (with correct English grammar).

Line 20: The description of the multifactorial can leave out the levels of all treatments and simply mention the number of levels and which factors were manipulated, possibly adding a sample number. Without sufficient description, words like "regreen" and "anthesis" are good parameters but difficult to understand out of context.

Line 46: "Nowadays" is too colloquial, and "have been considered as the most constrains limiting plant survival" has at least three instances of incorrect English grammar. The next sentence can be comprehended, but it also has numerous errors: "Moreover, the collaborative movement salts and water makes the water deficit and ineffectiveness, and threats to the sustainable yield of crop and forage and thus to the fragile ecosystems in the salt affected areas."

Line 51: "which is a C3" is an incomplete clause. "C3" refers to photosynthetic strategy.

Line 53: Please avoid colloquial contractions such as "it’s".

Line 54: "specie" must be "species", where "species" is both the singular and plural.

Line 56: A tilda (~) is used to signify an equivalence more approximate than an equals sign (=), unlike the dash (-), which is used to signify a range from one numerical value to another.

Line 58: "self-sufficient" is an adjective should be the noun "self-sufficiency".

Line 64: "There is huge gap" lacks the necessary article "a" and "huge" is a collquial adjective.

Line 67: "hard" is a collquilism; you should use "difficult".

Line 69: "maximum" is a noun; you should use the verb "maximize".


There are too many errors in English to continue my review.

Experimental design

There are too many errors in English to continue my review.

Validity of the findings

There are too many errors in English to continue my review.

Additional comments

There are too many errors in English to continue my review.

This could be a very good paper, given the impressive results of yield increase and your rigorous methods, but you must have someone sufficiently fluent in English revise this manuscript to become readable. Scientific papers need to be a permanent record, so readability is non-trivial. You do not have to use flowery language (e.g. beginning sentences with "thus" or "hence"), but you must minimize colloquial or ambiguous language. It is also very unusual to report ranges of values of measured physical quantities instead of means and standard deviations, which more accurately communicate central tendencies and errors to future readers.

·

Basic reporting

The manuscript addresses key issues (water scarcity and soil salinity) that are currently challenging agriculture around the world. However, the manuscript could be improved significantly with the help of a professional English editor or a native speaker. I strongly recommend assistance for English writing in order to convey this important research to a wider audience.
The literature references are supplied and relevant although more recent references could help to consolidate the old references. Adequate background to the problem in the region where the study was carried out was provided to put the study in context.
However, the introduction section can be revised and structured to improve the flow of ideas from presenting the context, identifying the problem and research gap and presenting the objectives.
The study addresses key issues and is relevant for modern agriculture and should be considered for publication provided the recommended revisions are attended to.

Experimental design

While water deficit and soil salinity have been widely investigated, the nature and complexity of plant and soil responses and their interactions with climate mean that local studies for different places still hold key answers to address these constraints. Therefore, the research fulfills the scope of this journal.
The research is well defined and relevant although the research gap was vaguely presented (possibly as a result of weak command of the English language). This can be rectified by soliciting professional help on language.
The methods and materials show that the study was rigorous enough and met ethical standards. The details provided in this section were enough to be replicated and I only suggested minor editorial editions.

Validity of the findings

The results are mostly valid and also adequately supported with literature. Minor revisions suggested include revisions of some arguments which do not change the structure of the manuscript entirely.
Only minor editions to the conclusions are suggested to align with objectives and research gap

Additional comments

Well done on carrying out the work for an adequate number of seasons.
However, please note that soil properties are usually difficult to deal with over short periods of time since most pedological processes are long term processes. They should be treated with caution.

---

## Round 0.2 · accepted · Accept

The authors made the efforts to address the reviewers' concerns. The manuscript is now ready for publication. Congratulations to the authors.

# # Reviewer 1 ·

Basic reporting

As I can see from the previous review, you have corrected all of the reviwers comments. It is ready for publication.

Experimental design

no

Validity of the findings

no

·

Basic reporting

No comment

Experimental design

No comment

Validity of the findings

No comment

Additional comments

All comments raised were attended to satisfactorily.